# Immunoprotective Analysis of the NFA49590 Protein from *Nocardia farcinica* IFM 10152 Demonstrates Its Potential as a Vaccine Candidate

**DOI:** 10.3390/pathogens11121488

**Published:** 2022-12-07

**Authors:** Lichao Han, Xingzhao Ji, Caixin Yang, Shuo Zhao, Shihong Fan, Lijun Zhao, Xiaotong Qiu, Jiang Yao, Xueping Liu, Fang Li, Zhenjun Li

**Affiliations:** 1State Key Laboratory of Infectious Disease Prevention and Control, National Institute for Communicable Disease Control and Prevention, Chinese Center for Disease Control and Prevention, Beijing 102206, China; 2Department of Pulmonary and Critical Care Medicine, Shandong Provincial Hospital Affiliated to Shandong First Medical University, Jinan 250021, China; 3Department of Epidemiology, School of Public Health, Shanxi Medical University, Taiyuan 030001, China; 4Sericulture and Apiculture Research Institute, Yunnan Academy of Agricultural Science, Mengzi 661100, China; 5Department of Laboratory, School of Laboratory Medicine and Life Sciences, Wenzhou Medical University, Wenzhou 325035, China; 6Department of Medicine, Tibet University, Lhasa 850001, China

**Keywords:** *Nocardia farcinica*, NFA49590, antigenicity, immunoprotection, vaccine

## Abstract

*Nocardia* is emerging as a serious and easily neglected pathogen in clinical practice with multidrug resistance that extends the treatment period for months or even years. This has led to the investigation of a vaccine approach to prevent *Nocardia* infections. However, studies on the protective proteins of *Nocardia* have not yet been carried out. In the present work, over 500 proteins in the supernatant of *N. farcinica* IFM10152 were identified by LC–MS/MS. In silico analysis of these proteins with a high content (score > 2000) predicted that NFA49590 was one of the conserved proteins in *N. farcinica* strains with potential antigenicity. After the rNFA49590 protein was cloned and expressed in *E. coli* (DE3) and purified using a Ni-NTA column, its good antigenicity was confirmed with sera from mice immunized with different *Nocardia* species by Western blot. Then we confirmed its ability to activate innate immunity by examining the phosphorylation status of ERK1/2, JNK, p38, and p65 and the cytokine levels of IL-6, TNF-α, and IL-10. Finally, we evaluated its immunoprotective effect in BALB/c mice, and we found that mice immunized with rNFA49590 protein exhibited high antibody titers, enhanced bacterial clearance ability, and generated robust protective effects from the *N. farcinica* challenge. These results offer strong support for the use of NFA49590 protein as a vaccine candidate and open the possibilities for the exploration of a large array of immunoprotective proteins.

## 1. Introduction

*Nocardia* are Gram-positive, partially acid-fast bacilli that can be found in soil, decaying vegetation, fecal deposits, and fresh and saltwater [1,2]. As an opportunistic pathogen, it has emerged as a threatening cause of pneumonia, brain abscess, primary cutaneous disease, purulent pericarditis, mediastinitis, bacteremia, and ocular nocardiosis in hospitalized patients over the last few decades [1,2]. It typically occurs in patients with depressed cell-mediated immunity conditions, such as those with lymphoma, malignancies, HIV infection, diabetes, solid-organ transplant, and those receiving long-term treatment with steroids, but infection occasionally occurs in immunocompetent hosts as well [1,2,3,4,5].

General treatment recommendations for nocardiosis rely heavily on antimicrobial treatment based on antimicrobial susceptibility patterns, given that *Nocardia* species exhibit different types of intrinsic multiple drug resistance [1,6]. However, as antibiotic therapies for nocardiosis can extend for months or even years, resistance to first-line antibiotics has emerged in several areas [6,7,8]. The *rox* gene [9] of *N. farcinica* encodes a rifampicin monooxygenase that converts rifampicin to a new compound responsible for a marked reduction in antibiotic activity. Mutations at position 1408 of the chromosomal 16S rRNA gene [10] resulted in high-level aminoglycoside resistance in *N. farcinica* isolates. In addition, resistance to trimethoprim-sulfamethoxazole (TMP-SMX) in *N. nova* and *N. cyriacigeorgica* [11] was related to two enzymes of the folate biosynthesis pathway: dihydrofolate reductase (DHFR) and dihydropteroate synthase (DHPS). The extensive drug resistance of this pathogen warrants the development of immunological insights into treatment options against *Nocardia* infections.

After the first genome sequence of *N. farcinica* IFM10152 was completed in 2004 [12], some efforts have been made to investigate gene expression related to drug resistance, virulence, and secondary metabolism. Mce1E [13] was proven functionally similar to *Mycobacterium tuberculosis* Mce proteins, which enable mammalian cell invasion and, thus, pathogenesis, leading to long-term survival in host cells. NFA34810 [14] was proven to facilitate *N*. *farcinica* invasion of host cells, and deletion of the *nfa34810* gene attenuated the invasion ability of host cells. However, few attempts have been made to investigate the immunoprotective proteins from *N*. *farcinica* thus far.

To establish effective pathogen–host interactions, pathogens typically exert their functions by transporting a number of effector proteins to the bacterial surface or the extracellular environment. Secreted proteins are directly involved in the communication between pathogens and hosts and are important components in the pathogen’s pathogenic and immunogenic effects. Identification and characterization of these proteins produced by *N*. *farcinica* are major advances in the understanding of the pathogenesis of *Nocardia* infections and have therefore opened promising avenues for the development of vaccines against this pathogen. In the present study, over 500 secreted proteins of *N. farcinica* IFM10152 were identified by LC–MS/MS. In silico analysis indicated that NFA49590 was one of the conserved proteins in *N. farcinica* strains with potential antigenicity. Then, the rNFA49590 protein was expressed, and its antigenicity was detected. By analyzing its immunoprotective efficacy in in vitro and in vivo experiments, we concluded that rNFA49590 protein elicits a potent protective response and holds great promise as an improved antigen for use in future vaccine formulations.

## 2. Materials and Methods

### 2.1. Animals and Ethics Statement

Wild-type female BALB/c mice (6–8 weeks of age) were purchased from SPF (Beijing) Biotechnology Co., Ltd. (Beijing, China). All mice were bred under specific pathogen-free conditions according to the guidelines. All animal experiments and procedures were approved by the Ethics Review Committee of the National Institute for Communicable Disease Control and Prevention at the Chinese Center for Disease Control and Prevention.

### 2.2. Bacterial Strains, Plasmid, and Cells

The standard bacterial strains used in this study included *N. farcinica* IFM10152, *N. asteroids* DSM43757T, *N. cyriaciegeorgoca* DSM44484T, *N. brasiliensis* DSM43758T, *N. otitidiscaviarum* DSM43242T, *N. transvalensis* DSM43405T, *N. veterana* DSM44445T, and *N. nova* DSM44481T. All the *Nocardia* strains were procured from the German Resource Centre for Biological Materials and cultured in BHI medium (Oxoid Ltd., Hants, UK) at a 37 °C incubator. *Escherichia coli* BL21(DE3) cells were procured from TransGen Biotech and cultured in an LB medium containing 50 µg/mL kanamycin. The pET30a plasmid was constructed in our laboratory and used to express *N. farcinica* NFA49590 in *E. coli*. The mouse cell line RAW264.7 (National Infrastructure of Cell Line Resource, Beijing, China) was cultured in DMEM medium supplemented with 10% fetal bovine serum (FBS).

### 2.3. LC–MS/MS and In Silico Analysis

After *N. farcinica* IFM10152 was cultured to exponential phase, the supernatant containing proteins was harvested and precipitated with 10% (*w*/*v*) TCA at 4 °C overnight. The precipitates were collected by centrifugation for 10 min at 14,000 rpm, washed three times with ice-cold acetone, dried, and stored at −20 °C before being analyzed using a TripleTOF 5600+ (SCIEX, Mundelein, IL, USA). After separation by an Eksigent microLC 415 system with a microliter flow rate, the proteins were identified by the search engine Mascot V2.3 (Matrix Science Ltd., London, UK) with a 95% or higher confidence interval of their scores. Finally, we screened one of the proteins (score > 2000) for conservative analysis using protein BLAST in the NCBI database (https://www.ncbi.nlm.nih.gov/, accessed on 25 March 2020) [15] and searching (I) subcellular localization using PSORTb (https://www.psort.org/psortb/, accessed on 25 March 2020) [16], (II) number of transmembrane helices using HMMTOP (http://www.enzim.hu/hmmtop/, accessed on 25 March 2020) [17], (III) signal peptides and gene function using UniProt (https://www.uniprot.org/, accessed on 25 March 2020) [18], and (IV) antigenic propensity using Protein Variability Server (http://imed.med.ucm.es/PVS/, accessed on 25 March 2020) [19].

### 2.4. Preparation of Recombinant NFA49590 Protein

The *nfa49590* gene was PCR-amplified using *N. farcinica* IFM10152 genomic DNA with the following primers: forward 5’-ACATGAATTCATGGTCGAGGTCGACTGT-3’ and reverse 5’-ACATAAGCTTTCAGCCGATGCTGAACGG-3’. The PCR product (714 bp) was digested by *EcoR I* and *Hind III*, then ligated into pET-30a(+) to generate pET30a-*nfa49590*. Subsequently, the recombinant plasmid was sequenced and then transformed into *E. coli* BL21 by electroporation. Transformants were selected on LB agar plates and confirmed by PCR.

Recombinant *E. coli* BL21 cells were cultured in an LB medium containing 50 µg/mL kanamycin at 37 °C. Isopropyl β-D-1-thiogalactopyranoside (IPTG; 0.2, 0.5, and 1 mM) was added when the OD_600_ reached 0.8, and protein expression was induced overnight at 16 °C, or 4 h at 28, 37 °C, respectively. After the bacterial suspension was sonicated and centrifuged (12,000 rpm/min, 4 °C for 20 min), both the pellet and culture supernatant were analyzed by SDS–PAGE. The recombinant NFA49590 protein (rNFA49590) was then purified using the His-Bind purification kit (Novagen, Germany). The collected protein was denatured and dissolved using 6 M urea, and the supernatant was filtered by a 0.45 µm millipore filter and loaded onto a Ni-NTA column equilibrated with an equilibration buffer. Nonspecific proteins were removed by washing with five-column volumes of imidazole buffer (20, 40, 60, 100, and 250 mM). All eluted fractions were collected and analyzed by SDS–PAGE. Purified rNFA49590 proteins were then refolded in the dialysate. Endotoxin in the purified rNFA49590 preparations was removed using a ToxinEraser endotoxin removal kit (GenScript, Nanjing, China) according to the manufacturer’s guidelines. The concentration of rNFA49590 was determined by a bicinchoninic acid (BCA) assay (Tiangen Biotechnology, Beijing, China), and the protein was stored at −80 °C.

### 2.5. Preparation of rNFA49590 and Whole Bacteria Antiserum

The concentration of the recombinant protein was adjusted to 400 µg/mL in PBS and mixed 1:1 (*v*/*v*) with aluminum hydroxide adjuvant (Bioss, Beijing, China). Then, the antiserum was prepared by subcutaneously immunizing mice with 100 µL protein–adjuvant mixture (10 µg rNFA49590 per mouse). Booster doses were given with the same agent on the 14th and 28th days.

For the preparation of polyclonal antibody serum, each *Nocardia* strain in the exponential growth phase was resuspended in PBS. Mice were infected by subcutaneous multipoint injection of 100 µL (2 × 10^7^ CFU) of each bacterial suspension three times every 2 weeks. All sera were collected 7 days after the last infection. In addition, mice were infected intranasally with 50 µL (1 × 10^7^ CFU) of *N. farcinica* IFM10152 under anesthesia. Sera were collected 3 and 7 days postinfection, and antiserum titers were determined by ELISA. Mouse antisera infected with *Mycobacterium bovis* were gifted by Dr. Xiuli Luan, Branch of Tuberculosis, Chinese Center for Disease Control and Prevention.

### 2.6. Subcellular Localization

Each subcellular fraction of *N. farcinica* IFM10152 was isolated as follows: After centrifugation of the cultured bacterial suspension, the supernatant and cell pellet were obtained separately. Methanol and trichloromethane were added to the supernatant and then centrifuged at 15,000 rpm and 4 °C for 5 min. The cell pellet was then washed twice with methanol and resuspended in PBS to obtain the secreted protein. The precipitate obtained in the first step was resuspended in PBS and lysed by sonication supplemented with a protease inhibitor. After cell debris and nonlysed cells were removed by centrifugation at 3000× *g* and 4 °C for 5 min, the supernatant was subjected to ultracentrifugation (27,000× *g*, 4 °C for 30 min) to separate the cell membrane and cytosolic fractions. Then, equal amounts of protein from each fraction were analyzed by Western blotting using anti-rNFA49590 serum as the primary antibody.

### 2.7. Antigenicity Determination by Western Blot

rNFA49590 proteins were separated by SDS–PAGE (5–12%) and transferred onto polyvinylidene fluoride (PVDF; Merck, Germany) membranes at 15 V for 1 h. Subsequently, the membranes were blocked with 5% skim milk in PBST at 4 °C for 2 h. To confirm recombinant protein expression and antibody production during infection, horseradish peroxidase (HRP)-conjugated monoclonal anti-pentahistidine (His) antibody (1:4000; New England Biolabs Inc., Ipswich, MA, USA), anti-rNFA49590 sera (1:2000), and anti-*N. farcinica* IFM10152 sera (1:2000) from mice were used as the primary antibodies. To analyze the specificity of the rNFA49590 protein, 1:2000 dilutions of antisera from *N. asteroids*, *N. cyriaciegeorgoca*, *N. brasiliensis*, *N. otitidiscaviarum*, *N. transvalensis*, *N. veterana*, *N. nova*, or *M. bovis* were used as the primary antibodies, and HRP-conjugated goat anti-mouse IgG (1:4000; SouthernBiotech, Birmingham, AL, USA) antibody was used as the secondary antibody.

### 2.8. Mitogen-Activated Protein Kinase (MAPK) and NF-κB Analysis

For MAPK and NF-κB analysis, RAW264.7 cells were seeded in 6-well microplates at a density of 8 × 10^5^ cells per well for 16–18 h. To exclude the effects of residual LPS in rNFA49590 protein, the preparation was preincubated with 100 ug/mL polymyxin B (PmB, a specific inhibitor for LPS, INALCO, USA) at 37 °C for 2 h. Then 2, 4, or 8 μg/mL of rNFA49590 protein (with or without 100μg/mL PmB) or 100 ng/mL LPS (with or without 100μg/mL PmB) was added to the cell plate. At the 30, 60, and 120 min time points, whole-cell extracts were harvested using RIPA lysis buffer (strong) (CWBIO, Beijing, China) containing 1% protease and 1% phosphatase inhibitor cocktail. After the protein concentration was measured using a BCA protein assay kit, equal amounts of protein were separated by sodium dodecyl sulfate polyacrylamide gel electrophoresis (SDS–PAGE) and transferred to PVDF membranes as described before. Primary antibodies against p-ERK1/2 (1:1000, CST, Danvers, MA, USA), p-JNK (1:1000, CST, Danvers, MA, USA), p-P38 (1:1000, CST, Danvers, MA, USA), p-P65 (1:1000, CST, Danvers, MA, USA), and β-actin (1:4000, CST, Danvers, MA, USA) were used. Secondary antibodies included HRP-conjugated goat anti-rabbit IgG (1:1000, Beyotime, Shanghai, China) or HRP-conjugated goat anti-mouse IgG (1:4000, ZSGB-BIO, Beijing, China).

### 2.9. Cytokine Measurements

For the measurement of cytokines, RAW264.7 cells were seeded at a density of 2 × 10^5^ cells per well in 24-well microplates and stimulated with 2 μg of rNFA49590 (with or without 100μg/mL PmB) or 100 ng/mL LPS (with or without 100μg/mL PmB). Culture supernatants were then harvested at the indicated times. To block MAPK and NF-κB signaling, cells were pretreated for 1 h with 20 µM specific inhibitors of p38 (203580, Sigma, Louis, MO, USA), ERK (PD 98059, Sigma, Louis, MO, USA), JNK (SP 600125, Sigma, Louis, MO, USA) or P65 (BAY11-7082, Sigma, Louis, MO, USA) prior to rNFA49590 protein exposure. The cytokine concentrations were determined by IL-6, TNF-α, and IL-10 ELISA kits (BD OptEIA™, San Diego, CA, USA) under the manufacturer’s instructions.

### 2.10. Mouse Immunization

Female mice were randomly divided into rNFA49590 (n = 26) and PBS (n = 26) groups and immunized with rNFA49590 protein or PBS three times, as mentioned above. Whole blood (with or without the addition of the anticoagulant heparin) from rNFA49590- and PBS-immunized mice (n = 6, respectively) was collected 7 days after the last immunization. Sera were isolated from whole blood (without the addition of the anticoagulant heparin) for the determination of rNFA49590-specific IgG, IgG1, IgG2a, and IgG2b (Abcam, Cambridge, UK) antibodies by ELISA. 

### 2.11. Whole Blood and Neutrophil Killing Assay

Equal whole blood (with the addition of the anticoagulant heparin) from two groups was mixed with 1 × 10^6^ CFU *N. farcinica* suspension. After 2 h incubation at 37 °C, serial dilutions of mixtures were plated on BHI agar plates for CFU count. 

After blood collection, mice were sacrificed by cervical dislocation. Femurs and tibias were dissected in a sterile environment, and bone marrow cells were flushed out with Hank’s balanced salt solution (HBSS; Solarbio, Beijing, China) using a syringe and needle. Marrow suspensions were then aspirated repeatedly with a syringe, followed by centrifugation at 500× *g* for 10 min at 4 °C. The pellets were then resuspended with 45% Percoll and over-layered onto Percoll gradient with 81%, 62%, 55%, and 50% layers for centrifugation at 500× *g* for 30 min. Bone marrow neutrophils were collected from the 81%/62% interface and washed in RBC lysing buffer (BD OptEIA™, San Diego, CA, USA) for 5 min, then washed with HBSS twice. Cells were then resuspended in RMPI 1640 medium and incubated in a 24-well microplate at a density of 2 × 10^5^ cells per well. Subsequently, *N. farcinica* suspension was added to wells at a ratio of 10:1 for 2 h. For bacterial survival determination, serial dilutions of cell lysates were plated on BHI agar plates. After 48 h incubation at 37 °C, the colonies were counted.

### 2.12. Mouse Infection

For immunoprotective analysis, mice in the rNFA49590 (n = 10) and PBS (n = 10) groups were inoculated intranasally under anesthesia with 50 µL of *N. farcinica* IFM 10152 (1 × 10^7^ CFU) suspension. Mouse weight and body temperature were quantified immediately prior to *N. farcinica* infection and 24 h postinfection. Then, the mice were dissected in a sterile environment, pulmonary bronchoalveolar lavage fluid (BALF) was obtained through 3–5 successive lavages of the bronchi with 1 mL of ice-cold PBS, and lactate dehydrogenase (LDH) in BALF was determined using the LDH-Glo™ Cytotoxicity Assay (Promega, Madison, WI, USA) following the manufacturer’s instructions. Whole lungs were collected and homogenized in 1 mL of PBS, and serial dilutions of homogenate were plated on BHI agar plates for bacterial counts. Lung homogenate was then centrifuged to obtain the supernatant, and the levels of TNF-α, IL-10, and IFN-γ were determined by ELISA.

For survival analysis, mice in the rNFA49590 (n = 10) and PBS (n = 10) groups were intraperitoneally administered 100 µL of an *N. farcinica* (5 × 10^9^ CFU) suspension. Then, mouse survival was monitored daily for a 10-day period. To assess overall tissue pathology, lung, liver, and kidney were dissected from surviving mice and fixed in 4% paraformaldehyde for 24 h, then embedded in paraffin and sectioned (5 mm). Slides were stained using hematoxylin and eosin (H&E) and then imaged with a biological microscope (Nikon, Eclipse Ci-L, Tokyo, Japan) according to the manufacturer’s instructions.

### 2.13. Statistical Analysis

All statistical analyses were performed using GraphPad Prism software version 9.0.0. Statistical differences were analyzed using ordinary one-way analysis of variance (ANOVA) with Tukey’s multiple comparisons. Survival rates were analyzed with the Kaplan–Meier method and the log rank test. For all tests, differences were considered statistically significant if the *p* value was less than 0.05.

## 3. Results

### 3.1. nfa49590 Was Conserved in N. farcinica Strains with Potential Antigenicity

Based on LC–MS/MS, over 500 proteins (score 15 to 8485) annotated as uncharacterized proteins, chaperone proteins, and putative esterases, proteases, oxidoreductases, etc., were identified (Appendix A). Table 1 shows that NFA49590 was one of the secreted proteins with high content in *N. farcinica* IFM10152 supernatants. Protein BLAST analysis indicated it was conserved in *N. farcinica* strains. In silico analysis revealed that NFA49590 is an extracellular protein with 237 amino acids and functions as MspA family porin (potential virulence factor). It possesses a signal peptide with 49 amino acids (1 aa~49 aa) and no transmembrane helix. Protein Variability Server predicted that NFA49590 possesses seven possible antigenic determinants (aa9~aa27, aa36~aa54, aa59~aa65, aa72~aa80, aa82~aa88, aa118~aa129, aa153~aa209) with an average antigenic propensity index of 1.0253, indicating its potential strong antigenicity.

### 3.2. Expression and Purification of rNFA49590 Protein

After detecting the PCR products by 2% agarose gel electrophoresis analysis, we obtained a single band of the target gene with the expected size (Figure 1A). DNA sequencing confirmed that the pET30a-*nfa49590* recombinant expression vector was constructed successfully. SDS–PAGE analysis indicated that rNFA49590 protein was mainly expressed in the form of inclusion bodies (Figure 1B). Protein expression was induced by 28 °C and 0.5 mM IPTG, and most of the heteroproteins were removed after 60 mM imidazole elution (Figure 1B,C). Purified rNFA49590 protein is presented as a single band (Figure 1D). rNFA49590 protein was then detected by BCA at a concentration of 950 μg/L

### 3.3. Subcellular Localization of Native NFA49590 Protein

To identify the localization of the native NFA49590 protein in *N. farcinica*, we isolated the subcellular components of *N. farcinica*. Western blotting revealed that native NFA49590 protein was mainly located in the culture supernatant (Figure 1E), indicating that it is a secreted protein.

### 3.4. Antigenicity of rNFA49590 Protein

We obtained anti-rNFA49590 sera from immunized mice. As shown in Figure 2A, rNFA49590 protein was specifically identified by anti-rNFA49590 sera. Then, we prepared antisera from mice immunized with *N. farcinica*, and Western blot analysis revealed that rNFA49590 protein was strongly identified by anti-His antibody and anti-*N. farcinica* sera by subcutaneous immunization (Figure 2B). In addition, we also found a specific band for the rNFA49590 protein that reacted with anti-*N. farcinica* sera by nasal immunization for 7 and 14 days (Figure 2B). These results indicate the good antigenicity of rNFA49590. To assess the specificity of rNFA49590 with antiserum, we used antisera from mice immunized with other *Nocardia* species and *M. bovis*. The results showed that the rNFA49590 protein could be recognized by anti-*N. asteroids*, anti-*N. cyriacigeorgica*, anti-*N. brasiliensis*, anti-*N. otitidiscaviarum*, anti-*N. transvalensis*, anti-*N. veterana*, and anti-*N. nova* sera, but not *M. bovis* (Figure 2C), which indicates that the rNFA49590 protein can cross-react with other *Nocardia* species and possesses interspecies specificity.

### 3.5. rNFA49590 Activated the MAPK and NF-κB Pathways in RAW264.7 Cells

The innate immune response is the first line of defense of the host against invasion by pathogenic microorganisms. The MAPK and NF-κB signaling pathways play important roles in the regulation of innate immunity. To determine whether MAPK and NF-κB were involved in the innate immune responses activated by the rNFA49590 protein, we examined the phosphorylation statuses of ERK1/2, JNK, p38, and p65 in RAW264.7 cells after stimulation with rNFA49590 (with or without 100μg/mL PmB) or 100 ng/mL LPS (with or without 100μg/mL PmB) by Western blotting. Then increased levels in p-JNK, p-JNK, p-p38, and p-P65 were observed at 30 min post-stimulation in rNFA49590, rNFA49590+PmB, and LPS-stimulated groups compared with the control group, but not in LPS+PmB group (Figure 3A), which indicated rNFA49590 protein capable of activating MAPK and NF-κB signaling pathways independently of residual LPS. Next, we stimulated RAW264.7 cells with 2 μg/mL of rNFA49590 protein for 30, 60, and 120 min, or 2, 4, or 8 μg/mL of rNFA49590 protein for 30 min. The results showed that the phosphorylation of ERK1/2, JNK, p38, and p65 occurred after stimulation with 2 μg/mL rNFA49590 for 30 min, and the phosphorylation level gradually decreased over time until it returned to baseline after 2 h of stimulation (Figure 3B). We also found that the phosphorylation levels of ERK1/2 and p65 gradually decreased with increasing protein concentration, while JNK and p38 gradually increased after 1 h of stimulation (Figure 3C), further indicating that rNFA49590 protein can activate the MAPK and NF-κB pathways in RAW264.7 cells in a dose- and time-dependent manner.

### 3.6. rNFA49590 Stimulated Cytokine Secretion through Activation of the MAPK and NF-κB Pathways in RAW264.7 Cells

After determining that rNFA49590 can activate the MAPK and NF-κB pathways, we further investigated the expression levels of downstream cytokines. The results showed that after stimulating RAW264.7 macrophages with 2 μg/mL protein for 6, 18, and 24 h, the concentrations of IL-6, TNF-α, and IL-10 in the cell supernatant showed increasing trends over time and reached higher levels at 24 h (Figure 4A). After the addition of PmB, the levels of each cytokine in the rNFA49590+PmB group remained high but not in the LPS+PmB group, which indicated that rNFA49590 could stimulate cytokine secretion independently of the effects of residual LPS. To further investigate the roles of the MAPK and NF-κB pathways in rNFA49590 protein-induced cytokine production, RAW264.7 cells were prestimulated with molecular signaling inhibitors followed by rNFA49590 protein for 24 h. Our results demonstrated that the expression of each cytokine was significantly downregulated after exposure to these four inhibitors (Figure 4B), indicating that the secretion of IL-6, TNF-α, and IL-10 by rNFA49590 protein-stimulated RAW264.7 cells was attributed to the MAPK and NF-κB pathways.

### 3.7. rNFA49590 Induces a High Humoral Response in Mice

To verify whether rNFA49590 induces a high humoral response, the quantities of serum IgG, IgG1, IgG2a, and IgG2b subclass specific to rNFA49590 were determined by ELISA assays. As shown in Figure 5A, rNFA49590-immunized mice elicited significantly higher titer of specific IgG, IgG1, IgG2a, and IgG2b compared with the control group. In addition, a *p* value of 0.0021 was achieved when comparing the IgG1 and IgG2a subclass responses, with IgG1 being higher, which indicates that rNFA49590 immunization appears to drive the immune response toward a Th2 bias.

### 3.8. Enhanced Bacterial Clearance Ability of Whole Blood and Neutrophils in rNFA49590-Immunized Mice

Blood and neutrophils play a key role in the clearance of *N. farcinica* in the host. After whole blood (with the addition of the anticoagulant heparin) collection, mouse bone marrow neutrophils were isolated to evaluate bacterial clearance ability. We then found a significant reduction in *N. farcinica* survival in whole blood (Figure 5B) and neutrophils (Figure 5C) in the rNFA49590-immunized group.

### 3.9. rNFA49590 Protein Protects Mice against Challenge with N. farcinica

To further test whether rNFA49590 protein protects mice against *N. farcinica* infections, two groups of mice, previously immunized with PBS or rNFA49590, were first intranasally challenged with nonlethal *N. farcinica*. We then detected an increase in body temperature and a decrease in body weight in the PBS-immunized group at 24 h postinfection, whereas smaller changes occurred in the rNFA49590-immunized group (Figure 5D,E). We also found that the LDH levels in the alveolar lavage fluid (Figure 5F) and the bacterial load in lung tissue (Figure 5G) of rNFA49590-immunized mice were significantly lower than those of the control group, indicating that the severity of lung infection in the rNFA49590-immunized group was reduced. Moreover, the determination of cytokines in lung supernatant proved that rNFA49590 immunization was able to prevent the release of TNF-α and IL-10 (Figure 5H) and promote IFN-γ secretion, which facilitated bacteria clearance.

Subsequently, PBS- and rNFA49590-immunized mice were challenged with a lethal dose of *N. farcinica* intraperitoneally, and the results showed an improved survival rate in the rNFA49590-immunized group (80%) compared with the PBS-immunized group (30%). No additional deaths were observed until Day 6 (Figure 6A). One mouse was then randomly selected from the remaining surviving mice in two groups, and the lungs, liver, and kidney tissues were dissected for pathological analysis. As shown in Figure 6B, lung tissue in the PBS-immunized group showed a large area of mild alveolar wall thickening, with more granulocyte infiltration in the alveolar wall (black arrow) and inflammatory cells in the blood vessels (green arrow). In contrast, lung tissue of the rNFA49590-immunized group showed a small area of the alveolar wall with mild thickening and a small number of granulocytes infiltrating the alveolar wall (black arrow). We also found a small number of granulocytes with small focal infiltrations around the liver lobules, central vein, and confluent area (black arrow) in the protein-immunized group of mice. In the PBS-immunized group, however, in addition to the more lymphocytic and granulocytic small focal infiltrates (black arrow), we also found localized hepatic sinusoidal dilatation (blue arrow), extramedullary hematopoietic foci in the hepatic lobules (yellow arrow), and inflammatory cells in the vessels (green arrow). Furthermore, we found that kidneys in the PBS-immunized group also showed significant inflammation response, with visible multifocal foci of necrosis, a small amount of tubular necrosis, fragmented nuclei, and eosinophilic homogeneous cytoplasm disintegration (red arrow), accompanied by lymphocyte and granulocyte dot infiltration (black arrows). In contrast, no significant inflammatory cell infiltration was observed in the rNFA49590-immunized group. These results corroborate the notion that the rNFA49590 protein protects mice during *N. farcinica* infections.

## 4. Discussion

With the characteristics of widespread distribution and extensive drug resistance, *Nocardia* causes serious damage to livestock and fish aquaculture as well as human beings [1,2,3], and the development of protective antigens subsequently offers new treatment options against *Nocardia* infections, especially in the immunocompromised host. It was estimated that an incidence of 0.37 cases per 100,000 population in immunocompetent hosts [20]; however, in immunocompromised patients, a high prevalence of 400 to 2650 cases per 100,000 population was reported among organ-transplant recipients [21]. Even with TMP/SMX and other antibiotic combination treatments, the prognosis of disseminated nocardiosis remains unsatisfactory, with a high mortality rate of >85% in immunocompromised individuals [22]. However, the development and application of *Nocardia* vaccines are not yet satisfactory. As facultative intracellular parasites, *Nocardia* can avoid macrophage killing by producing alkyl hydroperoxide reductase, catalase, and superoxide dismutase to detoxify reactive oxygen species (ROS) [23,24]. This peculiarity makes the development of effective vaccines particularly challenging. Although inactivated vaccines [25], live vaccines [25], DNA vaccines [26], and subunit vaccines [27] have been used in fish aquaculture to prevent *N. seriolae* infection, human vaccines for *Nocardia* infection are not yet available.

The availability of the complete genomic sequence from *N*. *farcinica* IFM10152 and advances in bioinformatics have made it possible for researchers to explore new possibilities for vaccine candidate identification against *Nocardia* pathogens. Many secreted proteins possess immunoprotective effects and are often used as vaccine candidates [28]. Secreted aspartyl proteinase 2 (Sap2) protein [29] is one of the leading vaccine candidates identified from *Candida*, and vaccination of mice improves survival during infection. Type III secreted protein (TTSP) [30] vaccination is an effective strategy against *Escherichia coli* O157, leading to decreased shedding in calves. In this work, we analyzed the proteins in the supernatant of *N*. *farcinica* IFM10152 identified by LC–MS/MS (score > 2000) and selected NFA49590 protein for further in silico analysis. To verify its potential as a vaccine candidate, we first established a prokaryotic expression vector in *E. coli* BL21(DE3) using the pET30a plasmid. Sufficient protein was then obtained under the optimal expression conditions of 28 °C and 0.5 mM IPTG. Purified rNFA49590 protein was confirmed by SDS–PAGE analysis as a single band with an expected molecular size of 34 kDa, indicating the successful expression of rNFA49590 protein.

Further antigenicity analysis showed that rNFA49590 is not only recognized by anti-*N*. *farcinica* sera from mice by subcutaneous or nasal immunization but also reacted with other *Nocardia* species antisera, which indicated the immune cross-reaction of rNFA49590 protein between *Nocardia* species antisera. This differed from the specificity of NFA34810 [14], which can only be recognized with anti-*N*. *farcinica* sera, but not anti-*N*. *cyriacigeorgica* or anti-*N*. *brasiliensis* sera. To further illustrate the role of rNFA49590 protein in innate and adaptive immunity, we first demonstrated that stimulating RAW264.7 cells with rNFA49590 protein could significantly activate the MAPK and NF-κB pathways. The MAPK and NF-κB signaling pathways have been proven to be activated by *Nocardia* and play key roles in innate immunity by mediating the production of inflammatory cytokines and proinflammatory cytokines [31]. Subsequent experiments also demonstrated that rNFA49590 promoted the production of IL-6, TNF-α, and IL-10 in RAW264.7 cells, which depended on the phosphorylation and activation of ERK, JNK, P38, and P65.

After determining its role in innate immunity, we attempted to monitor the immunoprotective efficacy of rNFA49590 in mice. To this end, a deeper dissection of the correlative rNFA49590 immunization in vivo was explored. Mice were administered nonlethal *N*. *farcinica* through the respiratory tract after immunization, given that the lungs were the most common site of infection and colonization. Our results revealed that rNFA49590 immunization elicited a robust functional humoral response, resulting in insignificant physical changes, reduced lung infection, decreased bacterial colonization, and proinflammatory cytokines in the lung supernatant. Further results of lethal doses of *N*. *farcinica*-infected mice also demonstrated the increased survival rate and reduced organ damage in the rNFA49590-immunized group. This immunoprotective efficacy of the rNFA49590 protein indicated its potential as an eligible vaccine candidate. 

Taken together, the results of the present study provide over 500 secreted proteins from *N*. *farcinica* IFM10152 supernatants and demonstrate the immunoprotective effect of the rNFA49590 protein in mice. The results showed that immunization with rNFA49590 led to a significant protective response in mice, which was mainly characterized by decreased inflammation and an increased survival rate. The NFA49590 protein is the first proven immunoprotective protein from *N*. *farcinica*, an initial and crucial step in the development of protective vaccines against *N*. *farcinica* infection. Ongoing work in our laboratory is further exploring additional immunoprotective proteins and then applying them clinically as vaccine reagents.

## 5. Conclusions

NFA49590 protein, which is abundant in *N*. *farcinica* IFM10152 supernatants, is a secreted protein with a molecular weight of 24 kDa. It has good antigenicity and can be recognized by antisera from mice infected with multiple *Nocardia* species but not *M. bovis*, indicating its interspecies specificity. This may provide a valid target for the identification of *N*. *farcinica* in the following work. In addition, the rNFA49590 protein can not only activate the MAPK and NF-κB signaling pathways but also protects mice from *N*. *farcinica* infection. In conclusion, NFA49590 can be considered a promising vaccine candidate against *N*. *farcinica* infection.

## Figures and Tables

**Figure 1 pathogens-11-01488-f001:**
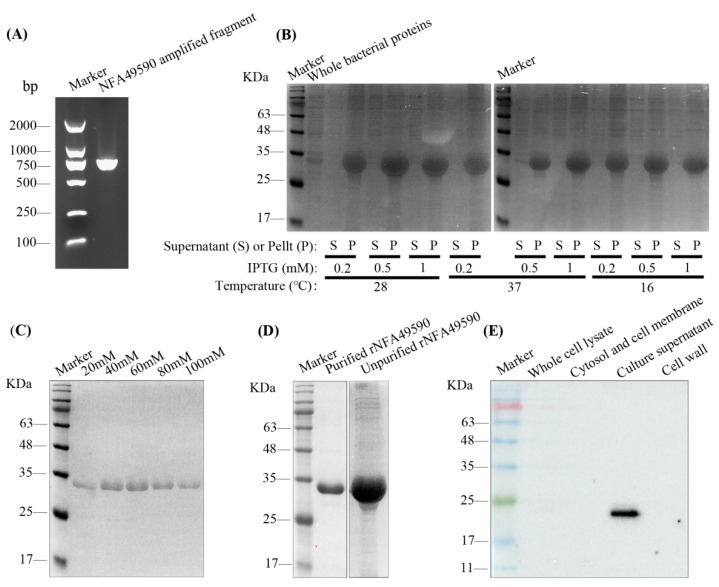
Expression, purification, and subcellular location of rNFA49590 protein. (**A**) Screening optimal expression conditions of location, IPTG concentration, temperature (**B**), and imidazole elution concentration (**C**) of rNFA49590 protein by SDS–PAGE. (**D**) Purified rNFA49590 protein. (**E**) Subcellular location of native NFA49590 protein.

**Figure 2 pathogens-11-01488-f002:**
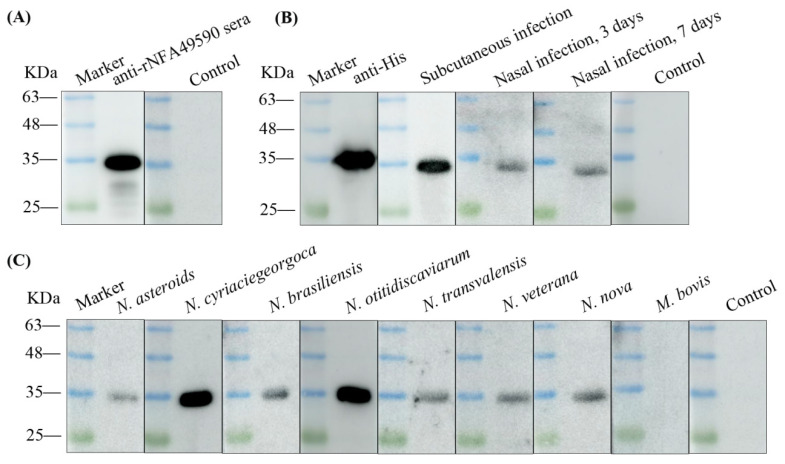
Reactivity of rNFA49590 protein with antiserum antibodies by Western blot. Recognition of rNFA49590 protein with anti-rNFA49590 sera (**A**) and antisera from mice infected with *N*. *farcinica* (**B**), other *Nocardia* species, or *M. bovis* (**C**).

**Figure 3 pathogens-11-01488-f003:**
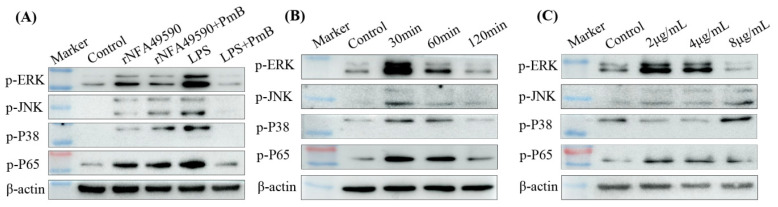
rNFA49590 protein activated the MAPK and NF-κB pathways in RAW264.7 cells. RAW264.7 cells were incubated with 2 μg/mL rNFA49590 (with or without 100 μg/mL PmB) or 100 ng/mL LPS (with or without 100 μg/mL PmB) for 30 min (**A**), or 2 μg/mL rNFA49590 protein for 30, 60, or 120 min (**B**), or 2, 4, or 8 μg/mL rNFA49590 protein for 30 min (**C**), and the phosphorylation statuses of ERK1/2, JNK, p38, and p65 were analyzed by Western blot. Data are representative of at least three independent experiments.

**Figure 4 pathogens-11-01488-f004:**
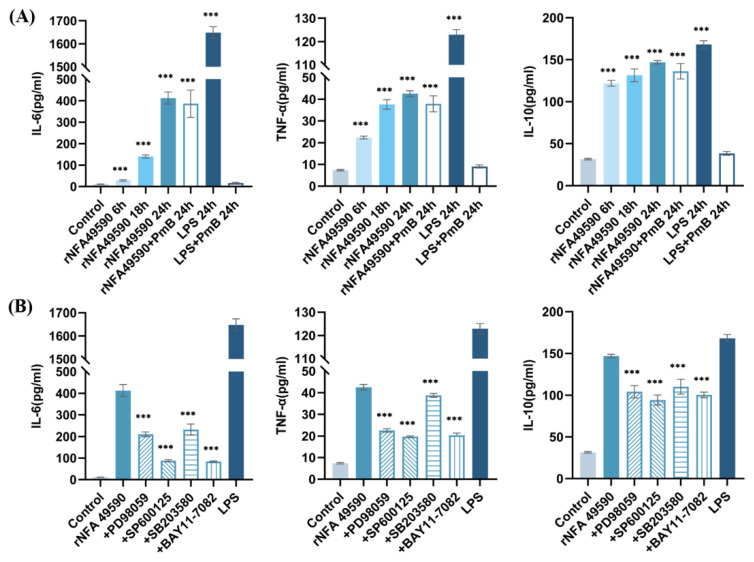
rNFA49590-mediated cytokine expression was dependent on the phosphorylation statuses of ERK1/2, JNK, p38, and p65. (**A**) RAW264.7 cells were incubated with 2 μg/mL rNFA49590 protein for 6, 18, and 24 h, or 2 μg/mL rNFA49590 protein (with 100 μg/mL PmB) or 100 ng/mL LPS (with or without 100 μg/mL PmB) for 24 h, and the levels of IL-6, TNF-α, and IL-10 in the supernatants were measured by ELISA. (**B**) RAW264.7 cells were pretreated for 1 h with inhibitors of 20 µM SB 203580, 20 µM PD 98059, 20 µM SP 600125, or 20 µM BAY11-7082 prior to rNFA49590 protein exposure, and cytokine levels were measured by ELISA. Data are expressed as the means ± SD for three independent experiments. *** *p* < 0.001 when compared with the control group (**A**) or rNFA49590 group (**B**).

**Figure 5 pathogens-11-01488-f005:**
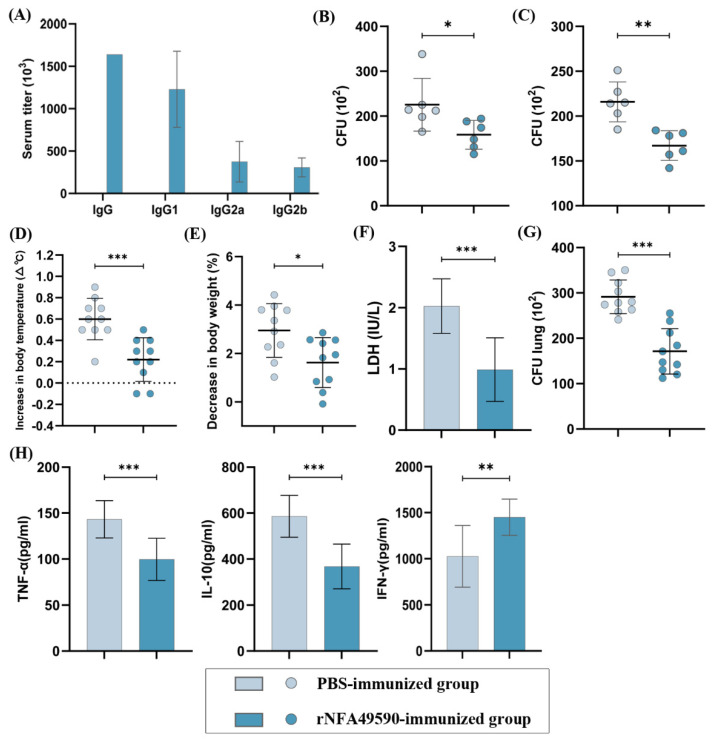
Effect of rNFA49590 protein immunization in mice. Female BALB/c mice were randomly divided into two groups and immunized with rNFA49590 protein or PBS three times. (**A**) Sera rNFA49590-specific IgG, IgG1, IgG2a, and IgG2b antibodies in PBS-immunized (n = 6) and rNFA49590-immunized (n = 6) mice were measured by ELISA. Bacterial survival in whole blood (**B**) and bone marrow neutrophils (**C**) after incubation for 2 h. (**D**–**H**) PBS-immunized (n = 10) and rNFA49590-immunized (n = 10) mice were intranasally infected with nonlethal *N*. *farcinica*, and then body temperature (**D**), body weight (**E**), LDH in BALF (**F**), CFU in lung tissue (**G**), and TNF-α, IL-10, IFN-γ in lung supernatant (**H**) were measured 24 h postinfection. Data are expressed as the means ± SD for three independent experiments. * *p* < 0.05, ** *p* < 0.01, *** *p* < 0.001.

**Figure 6 pathogens-11-01488-f006:**
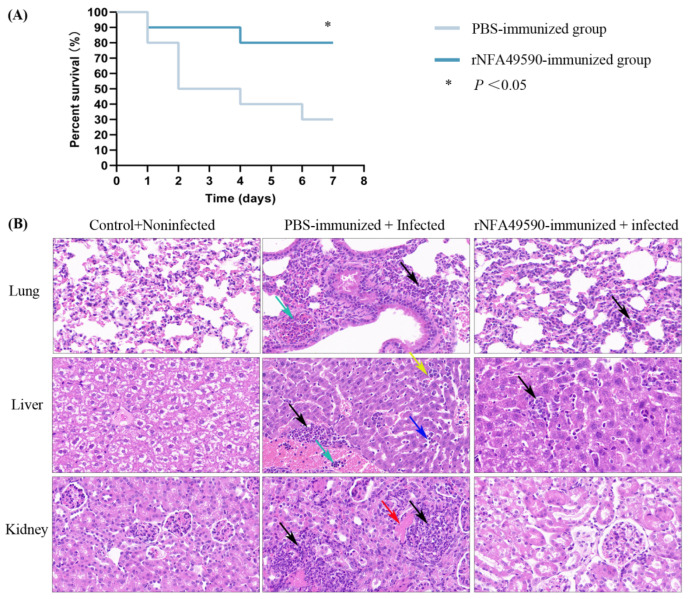
rNFA49590 immunization enhanced survival of mice after being challenged with *N. farcinica*. rNFA49590-immunized (n = 10) and PBS-immunized (n = 10) mice were challenged with a lethal dose of *N*. *farcinica* intraperitoneally, and mouse survival was monitored daily for a 10-day period (**A**). The remaining survival mice in two groups were sacrificed, and the lung, liver, and kidney (**B**) were dissected for histological examination (20×).

**Table 1 pathogens-11-01488-t001:** Secreted proteins in *N*. *farcinica* IFM10152 supernatants identified by LC–MS/MS (Score > 2000).

No.	Gene Code	Protein ID	Score	Matches	Protein Description
1	NFA49580	BAD59810	8485	376 (268)	hypothetical protein
2	NFA15900	BAD56436	7802	345 (252)	hypothetical protein
3	NFA56390	BAD60491	6316	320 (204)	hypothetical protein
4	NFA47630	BAD59615	4636	123 (96)	hypothetical protein
5	NFA1840	BAD55026	3856	208 (134)	putative esterase
**6**	**NFA49590**	**BAD59811**	**3686**	**181 (129)**	**hypothetical protein**
7	NFA54170	BAD60269	2837	100 (74)	putative protease
8	NFA54590	BAD60311	2622	131 (82)	hypothetical protein
9	NFA49620	BAD59814	2438	124 (75)	putative protease
10	NFA50960	BAD59948	2353	100 (84)	hypothetical protein

## Data Availability

Not applicable.

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
