# Peer review of "Immunoprotective Analysis of the NFA49590 Protein from Nocardia farcinica IFM 10152 Demonstrates Its Potential as a Vaccine Candidate"

_pathogens, 2022, doi:10.3390/pathogens11121488_

Round 1

Reviewer 1 Report

Han et al. report on the possible role of NFA49590 protein from Nocardia farcinica as immunogenic target for vaccine development. The study is adequately designed and described. Discussion can be improved, highlighting the importance of an effective vaccine in the setting of immunocompromised host - i.e. burden of breaktrough infections in subject in TMP/SMX prophylaxis, disease severity and impact on mortality. 

Overall, a thorough English revision is advisable.

Author Response

Han et al. report on the possible role of NFA49590 protein from Nocardia farcinica as immunogenic target for vaccine development. The study is adequately designed and described. Discussion can be improved, highlighting the importance of an effective vaccine in the setting of immunocompromised host - i.e. burden of breaktrough infections in subject in TMP/SMX prophylaxis, disease severity and impact on mortality. 

Overall, a thorough English revision is advisable.

Response: Thanks for your suggestions, and we agree with your recommendations. It was estimated an incidence of 0.37 cases per 100000 population in immunocompetent hosts, however, in immunocompromised patients, a high prevalence of 400 to 2650 cases per 100000 population was reported among organ-transplant recipients. Even with TMP/SMX and other antibiotic combination treatment, the prognosis of disseminated nocardiosis remains unsatisfactory, with a high mortality rate of >85% in immunocompromised individuals. We have supplemented it into the manuscript, Line 435-441. In addition, we have made some English corrections in the manuscript. Thank you again for your valuable comments.

Reviewer 2 Report

The current MS presents a well-performed study on the immunological response of an extracellular protein (NFA49590) derived from Nocardia farcinica IFM 10152. The authors could overexpress the protein in E. coli, the protein was produced insoluble and the authors could refold it and the purified form was used antigenic preparation. The authors could demonstrate both innate and adaptive immune activation of RAW264.7 cells as a result of rNFA49590 treatment. The protective effect of r rNFA49590  has also been proved.  Overall, the MS is well-organized and well-written, still, a few comments are to be addressed.

1-      I can’t find in the MS the reference to the protein function as has been annotated in the Uniprot, it should be a porin that contributes to the virulence of Nocardia.

2- Figure 3 shows a "ghost" bank in the marker lane. Also, there is splitting in the p-ERK bands, is this related to the antibody specificity?

3-      The methodology section can be revised to mention some details. For example, did the authors use antibiotics in the BHI agar plates as selective agents, is there further sample pre-treatment before the LC-MsMs experiment, what is the concentration of Urea solution used, etc? 

4-      It may be good to define abbreviations at least once in MS, for example, MAPK.

5-      Is it BALB/C or BABL/C?

Author Response

1-      I can’t find in the MS the reference to the protein function as has been annotated in the Uniprot, it should be a porin that contributes to the virulence of Nocardia.

Response: Thanks for your question. The function of nfa49590 is annotated as MspA family porin both in Uniprot and NCBI, which may be associated with the virulence of Nocardia. We have supplemented it into the manuscript, Line 262. Virulence factors are pathogenic and have strong immunogenicity. Many virulence proteins have been shown to exert protective effects after immunization to host, and are therefore selected as vaccine candidates. NFA49590 protein is highly conserved, strongly immunogenic, and a potential virulence factor, all of which confer favorable conditions as a protective antigen.

2- Figure 3 shows a "ghost" bank in the marker lane. Also, there is splitting in the p-ERK bands, is this related to the antibody specificity?

Response: Thanks for your question. I'm sorry that the Marker lanes were not clear in the process of converting the original color image to black and white. We have converted Figure 1 to Figure 3 to the original color images, and the molecular weight corresponding to each band of the Marker lane is shown in Figure 1E. In addition, the p-ERK molecule is composed of two bands (42, 44KDa), so the two bands shown in Figure 3 are normal and not a non-specific display of the antibody.

3-      The methodology section can be revised to mention some details. For example, did the authors use antibiotics in the BHI agar plates as selective agents, is there further sample pre-treatment before the LC-MsMs experiment, what is the concentration of Urea solution used, etc? 

Response: Thanks for your suggestion. The experiment was operated in class II biosafety cabinet, a strictly sterile environment, so we did not consider other bacteria contamination. And we did not find any contamination of other bacteria in BHI agar plates during the colony count, so we did not use antibiotics in the BHI agar plates. Before the LC-MS/MS experiment, the sample must be dried and stored at -20℃. And the concentration of Urea solution was 6M, we have added it in the manuscript, Line 129.

4-      It may be good to define abbreviations at least once in MS, for example, MAPK.

Response: Thanks for your suggestion. We have added the full names of some concept abbreviations, such as FBS (fetal bovine serum, Line 98), MAPK (mitogen-activated protein kinase, Line 177) and SDS-PAGE (sodium dodecyl sulfate polyacrylamide gel electrophoresis, Line 187 ), into the manuscript where the term first appears.

5-      Is it BALB/C or BABL/C?

Response: Thanks for your question. We misspelled BALB/c as BABL/c, and we have changed it to BALB/c in the manuscript. Thank you again for your valuable comments.

Reviewer 3 Report

The authors presented antigenicity of the Nocardia rNFA49590 protein. This protein can activate the MAPK and NF-κB pathways in monocyte/macrophage RAW264.7 cells, has anti-inflammatory activity, and induces a high humoral response in mice. The studies are significant. However, I would like to suggest some corrections:

1. Please explain why you studied Nocardia farcinica? N. farcinica is one of the least frequent clinically important species. Therefore, it would be more useful and expedient to study N. asteroides or N. brasiliensis.

2. In immunological response after vaccination, an important role have lymphocytes T, which were not studied in this paper. Authors research this topic indirectly by cytokines. I think should be studied these lymphocytes T here.

3. Authors presented that rNFA49590 can activate the NF-κB pathway. NF-κB has rather a role in the potentiation of vaccine adjuvants, not the main protein. Please, explain if you suggest that rNFA49590 can be adjuvant or main antigen?

4. Quality of Figure 6 is fatal. Please enlarge it, because neither lymphocytes nor granulocytes are visible.

5. Histology pictures of activated blood cells are missing, which would be necessary here.

Author Response

1. Please explain why you studied Nocardia farcinica? N. farcinica is one of the least frequent clinically important species. Therefore, it would be more useful and expedient to study N. asteroides or N. brasiliensis.

Response: Thanks for your question. The genus Nocardia currently contains more than 100 species, and clinically, the primary recognized human pathogens include N. farcinica, N. cyriacigeorgica, N. brasiliensis and N. asteroides. There are significant differences in the distribution of Nocardia species in different regions. For example, N. brasiliensis is more common in Mexico, however, N. farcinica and N. cyriacigeorgica are more common in China, France, etc. Hao et al. counted 441 Nocardia cases in China from 2009 to 2021, 176 of which belonged to N. farcinica, with the highest incidence rate of 39.9% [1]. D. Lebeaux et al. counted Nocardia cases in France between 2010-2015, and the most frequent species were N. farcinica (20.2%, 160/793) [2].

     On the other hand, N. farcinica is more virulent and lethal. For example, Nastaran Rafiei et al. counted 7 deaths in 20 cases with central nervous system infections, 6 of which were due to N. farcinica [3]. Considering the high clinical morbidity and high lethality, we selected N. farcinica as the study subject.

2. In immunological response after vaccination, an important role have lymphocytes T, which were not studied in this paper. Authors research this topic indirectly by cytokines. I think should be studied these lymphocytes T here.

Response: Thanks for your suggestion. We agree with you that the evaluation of lymphocytes T cell activation after immunization in mice is indeed a very important indicator. And there are some other important aspects that we need to explore and evaluate. In this study, we focused on evaluating the protective effect of humoral immunity induced by rNFA49590 protein in mice. And we found mice immunized with rNFA49590 protein exhibited high antibody titers, enhanced bacterial clearance ability, decreased inflammatory cytokine levels, and generated robust protective effects from N. farcinica challenge. Ongoing work in our laboratory is to evaluate the differences in the immunoprotective effects of rNFA49590 protein in mice of different sexes and ages, as we believe some differences exit in different hosts. Therefore, we sincerely hope elaborate this section in the next work, and it will be a comprehensive evaluation of cellular immunity (CD4 and CD8 T cells) and humoral immunity in mice of different sexes and ages. 

3. Authors presented that rNFA49590 can activate the NF-κB pathway. NF-κB has rather a role in the potentiation of vaccine adjuvants, not the main protein. Please, explain if you suggest that rNFA49590 can be adjuvant or main antigen?

Response: Thanks for your question. In this manuscript, we aim to demonstrate both innate and adaptive immune activation as a result of rNFA49590 treatment. The phosphorylation status of MAPK and NF-κB pathway in rNFA49590-treated RAW 264.7 cells were used to assess the ability of activating the innate immunity, not to illustrate its potential vaccine role. In this study, our results indicated that rNFA49590 protein was the main antigen, as we found the addition of rNFA49590 protein contribute the bacterial clearance both in whole blood and neutrophils. In addition, we found high-titers of serum specific IgG in rNFA49590 protein-immunized mice and exhibit a significant protective effect, while no specific IgG was detected in mice immunized with aluminum hydroxide adjuvant alone.

4. Quality of Figure 6 is fatal. Please enlarge it, because neither lymphocytes nor granulocytes are visible.

Response: Thanks for your valuable suggestion, that's important for us. We have enlarged Figure 6. And the lymphocytes and granulocytes were shown in Figure 6B.

5. Histology pictures of activated blood cells are missing, which would be necessary here.

Response: Thanks for your suggestion. In evaluating the immunoprotective effect of rNFA49590 protein in mice, we selected several perspectives such as survival rate, inflammatory response, bacterial clearance in lung tissue, and histopathological analysis of organs. We then found that rNFA49590-immunized mice had more granulocyte and lymphocyte infiltration in their organs compared to PBS-immunized mice, further corroborate the notion that rNFA49590 protein protects mice during N. farcinica infection. This is the first proven immunoprotective protein from N. farcinica, which can provide some insight into the development of the N. farcinica vaccines. In addition, there is still a lot of work that we need to do. Our laboratory is continuing to research other immunoprotective proteins of Nocardia, as well as the potential mechanism, in an effort to do more in-depth and comprehensive work in the development of Nocardia vaccine. Thank you again for your valuable suggestions, we have learned a lot from it.

[1] Wang H, Zhu Y, Cui Q, Wu W, Li G, Chen D, Xiang L, Qu J, Shi D, Lu B. Epidemiology and Antimicrobial Resistance Profiles of the Nocardia Species in China, 2009 to 2021. Microbiol Spectr. 2022 Apr 27;10(2):e0156021. doi: 10.1128/spectrum.01560-21.

[2] Lebeaux D, Bergeron E, Berthet J, Djadi-Prat J, Mouniée D, Boiron P, Lortholary O, Rodriguez-Nava V. Antibiotic susceptibility testing and species identification of Nocardia isolates: a retrospective analysis of data from a French expert laboratory, 2010-2015. Clin Microbiol Infect. 2019 Apr;25(4):489-495. doi: 10.1016/j.cmi.2018.06.013.

[3] Rafiei N, Peri AM, Righi E, Harris P, Paterson DL. Central nervous system nocardiosis in Queensland: A report of 20 cases and review of the literature. Medicine (Baltimore). 2016 Nov;95(46):e5255. doi: 10.1097/MD.0000000000005255.

Round 2

Reviewer 3 Report

The authors significantly corrected the manuscript according to the reviewer's suggestions. Recently, I recommend the article for publication.